# Updates and Perspectives on Aquaporin-2 and Water Balance Disorders

**DOI:** 10.3390/ijms222312950

**Published:** 2021-11-30

**Authors:** Yumi Noda, Sei Sasaki

**Affiliations:** 1Department of Nephrology, Nitobe Memorial Nakano General Hospital, Tokyo 164-8607, Japan; 2Department of Nephrology, Tokyo Medical and Dental University, Tokyo 113-8519, Japan; 3Department of Nephrology, Cellular and Structural Physiology Laboratory, Tokyo Medical and Dental University, Tokyo 113-8519, Japan; ssasaki.kid@tmd.ac.jp

**Keywords:** trafficking, diabetes insipidus, SIADH, congestive heart failure, hepatic cirrhosis, solute-free water diuretics

## Abstract

Ensuring the proper amount of water inside the body is essential for survival. One of the key factors in the maintenance of body water balance is water reabsorption in the collecting ducts of the kidney, a process that is regulated by aquaporin-2 (AQP2). AQP2 is a channel that is exclusively selective for water molecules and impermeable to ions or other small molecules. Impairments of AQP2 result in various water balance disorders, including nephrogenic diabetes insipidus (NDI), which is a disease characterized by a massive loss of water through the kidney and consequent severe dehydration. Dysregulation of AQP2 is also a cause of water retention with hyponatremia in heart failure, hepatic cirrhosis, and syndrome of inappropriate antidiuretic hormone secretion (SIADH). Antidiuretic hormone vasopressin is an upstream regulator of AQP2. Its binding to the vasopressin V2 receptor promotes AQP2 targeting to the apical membrane and thus enables water reabsorption. Tolvaptan, a vasopressin V2 receptor antagonist, is effective and widely used for water retention with hyponatremia. However, there are no studies showing improvement in hard outcomes or long-term prognosis. A possible reason is that vasopressin receptors have many downstream effects other than AQP2 function. It is expected that the development of drugs that directly target AQP2 may result in increased treatment specificity and effectiveness for water balance disorders. This review summarizes recent progress in studies of AQP2 and drug development challenges for water balance disorders.

## 1. Introduction

Maintaining water balance is essential for cell function and organism survival. The key event for its maintenance is water reabsorption in the collecting ducts, the terminal structure in the nephron [1]. This process is strictly regulated by the vasopressin-sensitive water channel aquaporin-2 (AQP2) [2,3,4,5,6,7,8]. AQP2 is abundant in the collecting duct and is largely stored in intracellular reservoirs of the collecting duct cells under conditions of normal hydration. When the body is dehydrated, vasopressin is secreted from the posterior pituitary gland. Circulating vasopressin then binds to the vasopressin V2 receptor on the basolateral membrane of the principal cells of the renal collecting duct and activates signal transductions promoting AQP2 translocation from the intracellular vesicles to the apical membrane, which enables water reabsorption from the urinary tubule. There are various water balance disorders caused by AQP2 impairment or dysregulation. This review summarizes recent findings of AQP2 studies and advances in targeting AQP2 in water balance disorders.

## 2. Upon Vasopressin Stimulation AQP2 Increases Water Reabsorption and Urine Concentration

Increases in body fluid tonicity and reductions in effective circulating blood volume stimulate the secretion of vasopressin from the posterior pituitary gland [1]. Circulating vasopressin then binds to the vasopressin V2 receptor on the basolateral membrane of the principal cells in the collecting duct and initiates intracellular signal transduction via coupling to heterotrimeric G-proteins [9,10]. Upon binding vasopressin, V2 receptors promote the disassembly of the heterotrimeric G-protein Gs, into Gsα and Gβγ subunits. The vasopressin-bound V2 receptor induces guanosine diphosphate/guanosine triphosphate (GDP/GTP) exchange, thereby activating the Gsα subunit to stimulate adenylate cyclase, which catalyzes a subsequent increase in the level of intracellular cyclic adenosine monophosphate (cAMP). cAMP activates protein kinase A (PKA), which then phosphorylates AQP2. Phosphorylated AQP2 translocates from the cytoplasm to the apical membrane, which renders the cell water permeable and results in water reabsorption [5]. Upon removal of the vasopressin stimulus, AQP2 is translocated back to the cytoplasm, which restores the water impermeability of the cell. Thus, vasopressin regulates the cycling of AQP2 between the apical membrane and the intracellular subapical storage vesicles of the collecting duct cells. Vasopressin increases water permeability by a factor of 10–100 in the collecting duct, inducing a rapid and drastic increase in water reabsorption [11].

Regulation of water transport by water channel translocation from the intracellular vesicles to the cell surface is also observed in other AQPs, including AQP1, AQP4, AQP5 [12,13,14,15]. As described above, the short-term effect of vasopressin is exerted by AQP2 translocation from the intracellular vesicles to the apical membrane. Furthermore, vasopressin can induce water reabsorption for 24 h or more by enhancing AQP2 transcription and its protein abundance in the collecting duct cells [16,17].

## 3. Phosphorylation Process of AQP2

PKA is a main regulator of AQP2 expression, phosphorylation and translocation to the apical membrane. Isobe et al. showed AQP2 expression was inhibited to an undetectable level in PKA knockout cells [18]. Moreover, vasopressin-induced AQP2 translocation to the apical membrane was impaired in PKA knockout cells overexpressing AQP2 [18]. Phosphorylation of serine 256 in AQP2 by PKA is important for AQP2 trafficking to the apical membrane [1,5,6]. Furthermore, there are three additional phosphorylation sites near the AQP2 C-terminus: serine 261, serine 264 and threonine 269 (or serine 269 in rodents). Yui et al. showed in rat AQP2 expressing MDCK cells that serine 256 phosphorylation, serine 269 phosphorylation and serine 261 dephosphorylation occur sequentially and that these three events are required for apical targeting of AQP2 [19]. Consistent with these findings, Sakai et al. show the phosphorylation profile using human urinary exosomes: phosphorylation of serine 256, 83%; phosphorylation of serine 261, 8%; phosphorylation of serine 264, 2%; phosphorylation of threonine 269T, 1% [20].

AQP2 translocation is also altered by other kinases than PKA. AQP2 serine 256 can also be phosphorylated by Golgi casein kinase. PKA-independent phosphorylation at serine 256 of AQP2 is increased during AQP2 transition through the Golgi apparatus, suggesting that phosphorylation by Golgi casein kinase may be required for Golgi transition [21]. Van Balkom et al. [22] showed that activation of protein kinase C induces AQP2 endocytosis from the apical membrane, which occurs independently of serine 256 phosphorylation. In addition, AQP2 exocytosis is shown to be altered by a cyclic guanosine monophosphate (cGMP)-dependent pathway [23], and an inhibitor of cGMP phosphodiesterase promotes AQP2 trafficking to the apical membrane [24].

Cheung et al. found that inhibition of Src, a non-receptor tyrosine kinase, leads to AQP2 phosphorylation at serine 269 and promotes AQP2 apical membrane accumulation independently of vasopressin signaling and serine 256 phosphorylation [25]. This finding represents a novel therapeutic target that could potentially be exploited to regulate water balance.

## 4. The Role of Calcium in the Regulation of AQP2

AQP2 trafficking is also regulated by intracellular Ca^2+^ mobilization [26]. In addition to increasing intracellular cAMP levels, vasopressin binding to V2 receptors stimulates a rapid increase of intracellular Ca^2+^, which is followed by sustained temporal oscillations of Ca^2+^ levels in the principal cells of the collecting duct. This process appears to be involved in AQP2 exocytosis. Balasubramanian et al. suggest several plausible candidates as downstream effectors of vasopressin-induced Ca^2+^ signaling, such as calmodulin and myosin light chain kinase [26]. Myosin light chain kinase was shown to be required for AQP2 trafficking, as described in the next section.

Calcineurin, a calcium-calmodulin-regulated serine-threonine phosphatase, and its downstream transcriptional effector NFATc (nuclear factor of activated T cell cytoplasmic) also regulate AQP2 [27]. Calcineurin dephosphorylates NFATc and promotes its nuclear translocation. Subsequently, NFATc binds to the promoter region of the *AQP2* gene and promotes AQP2 expression. On the other hand, Wnt5a is a ligand for frizzled receptors that increases intracellular calcium [28]. Ando et al. show that the Wnt5a/calcium/calmodulin/calcineurin signaling pathway induces AQP2 protein expression, phosphorylation and trafficking [29].

Calmodulin is also gaining attention in the study of intracellular translocation of the other aquaporin AQP4 [14,30,31,32]. AQP4 in astrocytes mediates water transport across the blood–brain barrier and plays an important role in central nervous system (CNS) edema. Calmodulin binds AQP4 and promotes AQP4 translocation from the intracellular vesicles to the cell surface. It is reported that inhibition of calmodulin with trifluoperazine inhibits AQP4 translocation, resulting in the prevention of CNS edema in a rat spinal cord injury model [14] and a stroke mouse model [30]. Calmodulin is also a promising drug target for CNS edema.

Extracellular Ca^2+^ is also involved in AQP2 regulation. Drug-induced hypercalcemia/hypercalciuria causes polyuria and reduces AQP2 expression in rats [33]. AQP2 translocation to the apical membrane prompted by forskolin-induced increases in cAMP levels is inhibited by increased levels of extracellular Ca^2+^ [34]. Furthermore, high luminal Ca^2+^ in the renal collecting duct attenuates vasopressin-induced AQP2 trafficking through calcium sensing receptor activation [35,36].

## 5. The Role of the Cytoskeleton in AQP2 Trafficking

The actin cytoskeleton is reported to function as a barrier for AQP2 exocytosis [37,38]. Actin depolymerization is necessary for the cAMP-dependent translocation of AQP2 [39,40]. In fact, stimulation of prostaglandin E3 receptors has been shown to inhibit vasopressin-induced inactivation of Rho GTPase, vasopressin-induced F-actin depolymerization, as well as AQP2 translocation induced by vasopressin, cAMP or forskolin [39]. Bradykinin-induced Rho GTPase activation stabilizes cortical F-actin and inhibits AQP2 trafficking [41].

The GTPase-activating protein Spa-1 (SPA-1) binds to the C-terminus of AQP2, which is required for AQP2 trafficking [42,43]. SPA-1 inhibits Rap1 GTPase-activating protein, which triggers F-actin disassembly and may maintain the basal mobility of AQP2 [8,44]. SPA-1-deficient mice show impaired AQP2 trafficking and hydronephrosis [42,45]. In humans, mutations in the SPA-1 binding region in the C-terminus of AQP2 cause nephrogenic diabetes insipidus (NDI), a disease characterized by a massive loss of water through the kidney [46,47].

Myosin II and its regulatory light chain are present in an AQP2-binding protein complex [48]. Myosin is reported to be critical for AQP2 recycling [49]. Myosin light chain kinase is a calmodulin-dependent kinase that regulates actin filament organization by phosphorylating the regulatory light chain of myosin. Chou et al. show that vasopressin induces myosin light chain phosphorylation [50]. Furthermore, myosin light chain kinase is required for vasopressin-induced actin depolymerization and AQP2 transition from early to late endosomes [51].

The actin-related protein (Arp)2/3 complex is a key factor in actin filament branching and polymerization. Inhibition of the Arp2/3 complex has been shown to prevent AQP2 exocytosis [52]. Using 3D super-resolution microscopy, Holst et al. reported that association of AQP2-containing vesicles with F-actin is enhanced by serine 256 phosphorylation [53].

## 6. AQP2 Recycling and Endocytosis

AQP2 is a recycling membrane protein. Upon vasopressin stimulation, AQP2 is transported to the apical membrane, rendering the cell water permeable as described above. After vasopressin stimulation is terminated, AQP2 is shuttled back to the cytoplasm, a process that restores the water impermeability of the cell. This recycling process occurs constitutively, and many signaling pathways are involved in the regulation of each stage of the recycling process. Vasopressin signaling is the most potent and most important factor that enhances the exocytotic process among the recycling processes.

During AQP2 endocytosis, AQP2 accumulates in clathrin-coated pits and is internalized via a clathrin-mediated mechanism [54,55]. Dynamin is a GTPase that is involved in the formation and pinching off of clathrin-coated pits to form clathrin-coated vesicles. GTPase-deficient dynamin mutants exhibit arrested endocytosis and accumulate AQP2 in the apical membrane independently of vasopressin stimulation [55,56].

The heat shock protein Hsc70 binds to AQP2 and is involved in AQP2 endocytosis [57]. Ubiquitination of lysine 270 of AQP2 is important for AQP2 endocytosis and degradation [58]. The E3 ubiquitin ligase CHIP interacts with AQP2, Hsp70, and Hsc70, and ubiquitinylates AQP2 [59]. CHIP knockdown increases AQP2 in the plasma membrane, indicating its involvement in AQP2 endocytosis and degradation [59].

## 7. The Molecular Mechanism Driving AQP2 Movement

As described above, AQP2 phosphorylation is required for its trafficking to the apical membrane. However, the mechanism by which this phosphorylation event induces the AQP2 movement was unknown until recently. In other words, the direct mechanism which generates motion in AQP2 trafficking was unknown. To clarify this mechanism, we attempted to identify AQP2-binding proteins and discovered that AQP2 is a component of a multiprotein motor complex [8,48,60,61].

To provide further insight into the intra-complex interactions crucial to AQP2 regulation, we applied fluorescence correlation spectroscopy (FCS) and fluorescence cross-correlation spectroscopy (FCCS) for the first time to channel research. As a result, we succeeded in measuring the spatial and temporal dynamics of the AQP2 motor complex components at the single-molecule level and discovered the direct mechanism that drives channel movement to the targeted site [5,37,38,62]. Under basal conditions, AQP2 binds to G-actin, while F-actin is stabilized by tropomyosin-5b (TM5b) to form a barrier that inhibits AQP2 translocation to the apical membrane. Vasopressin-triggered AQP2 phosphorylation releases AQP2 from G-actin and promotes AQP2 association with TM5b, which sequesters TM5b from F-actin and destabilizes the F-actin network, thereby allowing efficient movement of AQP2 to the apical membrane. This molecular mechanism was confirmed using purified recombinant proteins reconstituted in liposomes [5,37,38,62].

FCS and FCCS measurements are particularly powerful for clarifying the dynamics of multiprotein complexes at the single-molecule level and at various locations in the cell. Many channels and transporters form multiprotein complexes, and such intra-complex interactions are crucial in their regulation. Our methods, including FCS, FCCS, and a reconstituted purified protein system, are powerful techniques for investigating membrane proteins that form multiprotein complexes, clarifying related pathophysiology and identifying therapeutic targets for diseases involving membrane proteins.

## 8. The Water Channel Activity of Individual AQP2 Proteins

The effects of AQP2 phosphorylation on the water transport activity of individual AQP2 channels have also been extensively examined. Kuwahara et al. [63] examined the phosphorylation and osmotic water permeability (*P_f_*) of AQP2 expressed in *Xenopus* oocytes. cAMP stimulation increased the *P_f_* of oocytes expressing AQP2, which occurred in the absence of increased AQP2 levels on the oocyte surface, thereby suggesting that the *P_f_* of individual AQP2 channels was increased. Moeller et al. evaluated the *P_f_* and the plasma membrane abundance of wild-type (WT) and mutants of AQP2 expressing oocytes [64]. Both the *P_f_* and plasma membrane abundance of the S256A-AQP2 mutant (non-phosphorylation-mimick) were decreased compared with WT-AQP2, resulting in that *P_f_* values relative to the plasma membrane abundance were similar. This finding suggests that the absence of phosphorylation at this site have no effect on individual AQP2 protein function. However, the method used to determine plasma membrane abundance was semiquantitative, and this study could not exclude the possibility that the *P_f_* of individual AQP2 proteins was altered by this mutation. To quantitatively evaluate the *P_f_* of individual AQP2 proteins in the absence of the effects of other proteins, we examined the function of purified full-length recombinant human AQP2 reconstituted in liposomes [65]. This study provides direct evidence that the water transport activity of AQP2 is enhanced approximately 2-fold by phosphorylation at serine 256. In addition to AQP2 trafficking to the apical membrane, this study indicates that the water transport activity of individual AQP2 is involved in the regulation of water reabsorption from the urine in kidney collecting ducts.

Vasopressin increases water permeability of the collecting duct by a factor of 10–100, inducing a rapid and drastic increase in water reabsorption [11]. Thus, vasopressin-induced short-term regulation of *P_f_* of the collecting ducts appears to be mainly due to AQP2 translocation, with the altered water transport activity of individual AQP2 proteins acting to potentiate the effect.

## 9. The Role of AQP2 in Fluctuating Osmolality

The collecting duct cells where AQP2 is expressed, are exposed to great fluctuations in osmotic pressure during transitions between diuresis and antidiuresis, which are caused by AQP2-regulated water reabsorption. The promoter activity of the murine AQP2 gene is increased by hypertonicity and decreased by hypotonicity [66,67]. Acute hypertonicity induces AQP2 translocation to the apical membrane, while chronic hypertonicity induces AQP2 translocation to the basolateral membrane [68,69]. Cell volume regulation in response to external osmolality changes is a fundamental property of cells. When cells are exposed to hypotonic fluid, they swell because of osmotic water influx. After swelling, the cells start to recover their original volume. This defense process against hypotonic shock is called regulatory volume decrease (RVD) [70]. Hypotonicity induces AQP2 internalization, which may contribute to RVD by limiting water entry into cells [71]. Moreover, we discovered that AQP2 regulates cell volume decreases by controlling the cytoskeleton [72]. As well as being a water channel, AQP2 also contributes to the cell volume regulation of collecting duct cells.

## 10. NDI

Impairment of AQP2 results in NDI, which is characterized by an inability of the kidney to concentrate urine, even when the plasma concentrations of vasopressin are elevated. This condition results in a massive loss of water through the kidney, leading to severe dehydration. Unlike central diabetes insipidus, vasopressin administration is ineffective for patients with NDI. Moreover, there are congenital and acquired forms of NDI.

## 11. Congenital NDI

In 90% of cases of congenital NDI, the condition results from loss-of-function mutations in the AVPR2 gene encoding the vasopressin V2 receptor (X-linked NDI). The remaining cases result from mutations in AQP2 (autosomal NDI). AQP2 is located in 12q13.12. To date, 64 mutations in the AQP2 gene have been reported (The Human Gene Mutation Database; http://www.hgmd.cf.ac.uk/ac/index.php, accessed on 28 November 2021) (Figure 1; Ref. [6]). There are two inheritance types, autosomal-recessive and autosomal-dominant NDI. Most of the mutations in recessive NDI are located in the core region of the protein, which leads to misfolded proteins that become trapped in the endoplasmic reticulum and is degraded rapidly by the proteasome. On the other hand, AQP2 homotetramers composed only of wild-type proteins are properly translocated to the apical membrane, which explains the healthy phenotype of heterozygous individuals.

All mutations in autosomal-dominant NDI are located in the cytosolic C-terminus of AQP2. This region is essential for AQP2 translocation, with mutations impairing its translocation to the apical membrane, although the water channel function of these mutants is preserved [73,74,75,76]. In contrast to the AQP2 mutants in the recessive form of the disease, AQP2 mutants in the dominant form are not misfolded and able to form heterotetramers with WT-AQP2 and impair the apical targeting of heterotetramers composed of mutant and WT. This effect explains the dominant mode of inheritance with these mutations. Sohara et al. generated gene knockin mice with heterozygous mutant AQP2 resulting from a gene deletion (763–772del) that produces a mouse model of dominant NDI [77]. Mutant AQP2 is incorrectly translocated to the basolateral membrane, where it forms a heterotetramer with WT-AQP2 and shows a dominant-negative effect on the normal apical translocation of WT-AQP2. As a result, the urine concentrating ability of these mice is severely impaired.

## 12. Acquired NDI

Acquired NDI is more common than congenital NDI and is caused by a variety of conditions including drug treatments, electrolyte disturbances, and urinary tract obstruction. Dysregulation of AQP2 plays a crucial role in many acquired NDI.

Lithium is widely used in the treatment of bipolar disorder. Its most common adverse effect is NDI, which occurs in up to 40% of these patients using lithium [78]. In lithium-induced NDI, AQP2 expression and its apical targeting are both inhibited. Lithium enters cells expressing AQP2 via the epithelial sodium channel in the apical membrane and accumulates intracellularly. Lithium accumulation leads to the inhibition of signaling pathways that involve glycogen synthase kinase-3β (GSK3β). Rao et al. [79,80] showed that GSK3β inhibition by lithium results in increased cyclooxygenase 2 and reduced adenylyl cyclase activity, leading to the reduced cAMP generation and decreased AQP2 expression. Moreover, lithium reduces the proportion of principal cells in collecting ducts and increases the proportion of intercalated cells [81]. This restructuring of the collecting duct, together with down-regulation of AQP2, may be important in lithium-induced NDI.

Hypokalemia and hypercalcemia cause down-regulation of AQP2, which results in a vasopressin-resistant urinary concentrating defect. Autophagic degradation of AQP2 is involved in both hypokalemia and hypercalcemia-induced NDI [82,83]. In addition, hypokalemia alters the proportions of principal and intercalated cells as observed following lithium treatment [84].

Urinary tract obstruction is also a common cause of NDI and is associated with reduced AQP2 abundance. Recently, autophagic degradation of AQP2 has been shown to be involved in mediating this process [85].

## 13. Water Retention by AQP2 Dysregulation

AQP2 also plays a crucial role in the pathophysiology of water retention disorders. A well-known example of this is decompensated heart failure. Water retention and hyponatremia are common, clinically important complications of heart failure. Plasma vasopressin levels are suppressed by hyponatremia in healthy individuals; however, vasopressin levels are not suppressed in patients with heart failure and hyponatremia [86]. In patients with heart failure, a decrease in effective blood volume and atrial filling is sensed by the left atrial baroreceptors, resulting in stimulation of vasopressin secretion and the consequent increases AQP2 expression and trafficking to the apical membrane of principal cells of the collecting duct. In patients with heart failure, V2 receptor antagonists promote electrolyte-free water excretion and elevate serum sodium concentrations [87,88,89]. The vasopressin antagonist tolvaptan has been shown to improve several symptoms of heart failure, such as dyspnea, in these patients [90].

Water retention with hyponatremia is also a critical complication of hepatic cirrhosis. In these patients, nonosmotic secretion of vasopressin occurs as a secondary event to splanchnic arterial vasodilation and relative arterial underfilling [86]. Moreover, AQP2 expression was reported to be increased and correlated with ascites volume [91]. In patients with hyponatremic cirrhosis, tolvaptan raises plasma sodium levels and decreases in ascites, although these effects are limited to a short duration [92].

Syndrome of inappropriate antidiuretic hormone secretion (SIADH) is a disorder characterized by impaired water excretion caused by the inability to suppress vasopressin secretion. SIADH is the predominant cause of the commonly encountered disorder euvolemic hyponatremia [93]. However, antidiuresis is attenuated during conditions of chronic vasopressin excess, resulting in a degree of water diuresis. This has been referred to as the ‘vasopressin escape’ phenomenon [17]. Saito et al. [17,67] found that diminished AQP2 expression plays a role in preventing maximal urinary concentrating in SIADH model rats.

SIADH frequently occurs in association with vascular disease, infectious disease, or neoplasms in the lung or central nervous system. In patients with SIADH, the V2 receptor antagonist was shown to be effective in increasing urine volume and plasma sodium levels [94]. However, its long-term effect is limited in rats with SIADH [95]. Although AQP2 protein expression is reduced shortly after administration of the V2 receptor antagonist to rats with SIADH, expression subsequently increases in parallel with a decline in its therapeutic effects.

Urinary excretion of AQP2 is associated with vasopressin activity in the kidney and is a clinically useful biomarker [96,97]. AQP2 is excreted into the urine through the secretion of exosomes originating from intracellular vesicles of multivesicular bodies [98]. During this process, the outer membrane of multivesicular bodies fuses with the apical membrane. Urinary AQP2 excretion is increased by dehydration or vasopressin and decreased by hydration. Urinary AQP2 excretion is also increased in patients with heart failure, hepatic cirrhosis [99,100]. In patients with heart failure, administration of a V2 receptor antagonist produced a significant increase in urine flow and solute-free water excretion, accompanied with a drastic reduction in urinary AQP2 excretion [99,101]. Elevation of urinary AQP2 excretion is also observed in SIADH [102]. Urinary excretion of AQP2 is a sensitive marker of the antidiuretic activity of vasopressin. In addition, the pharmacological effect of tolvaptan can be monitored by urinary AQP2 levels in heart failure, hepatic cirrhosis and SIADH [101,103].

## 14. Development of Therapeutics for NDI by Targeting AQP2 Regulation

There is no cure for NDI, and it is currently managed by salt restriction combined with hydrochlorothiazide [6]. Hydrochlorothiazide reduces sodium reabsorption in the distal convoluted tubule, leading to increased sodium excretion and reduced extracellular fluid volume. As a result, the glomerular filtration rate decreases and proximal tubular sodium and water reabsorption increases. Consequently, less water and sodium are delivered to the collecting ducts, which results in decreased urine volume. However, current treatments do not sufficiently obviate the excessive water excretion. Therefore, extensive efforts to develop therapies are continuing.

As described above, cAMP is a major activator of AQP2. Several groups have examined ligands of G protein-coupled receptors (GPCR) that increase cAMP production as possible treatments for NDI. The GPCR ligand calcitonin increases cAMPand AQP2 trafficking in cultured cells, and urine osmolality during the first 12 h of treatment in vasopressin-deficient rats [104]. This effect of calcitonin is subsequently diminished over the following 72 h. The GPCR ligand secretin increases cAMP and AQP2 expression, but is not able to increase AQP2 trafficking to the apical membrane nor increase urine concentration in NDI model mice [105]. In this study, secretin plus fluvastatin was able to increase urine concentration. Renal tubule-specific EP4 knockout mice showed impaired urine concentrating defect [106]. Li et al. show that a selective ligand of the EP4 subtype of the prostaglandin E2 (PGE2) GPCR receptor elicits pronounced efficacy after the first day of infusion in attenuating polyuria in a mouse model of X-linked NDI that lacks V2R [107]. The effects of GPCR ligands on AQP2 and urine concentration are likely to be limited by receptor downregulation or desensitization. On the other hand, the E-prostanoid receptor (EP2) agonist butaprost induces a pronounced long-term response on AQP2 membrane targeting and urinary concentrating ability in rats [108,109].

The effect of cAMP phosphodiesterase on AQP2 trafficking has also been examined. Sohara et al. report that the PDE4 cAMP phosphodiesterase inhibitor rolipram increases cAMP content in the papillae, AQP2 phosphorylation, and apical membrane translocation of AQP2, resulting in increased urine osmolality in autosomal dominant NDI mice model [77].

AQP2 phosphorylation by cGMP kinase is also involved in its exocytosis, and the cGMP phosphodiesterase inhibitor sildenafil citrate induces AQP2 membrane insertion [23,24]. Therefore, cGMP phosphodiesterase inhibitors are expected to be effective in treating NDI due to V2R impairment by bypassing the requirement for cAMP signaling to produce AQP2 membrane insertion. However, a clinical trial showed that the PDE5-inhibitor sildenafil, or the soluble guanylate cyclase stimulator riociguat, increased cGMP levels but did not improve urinary concentration ability in patients with congenital NDI [110].

We showed the interaction of phosphorylated AQP2 with TM5b is essential for AQP2 trafficking to the apical membrane, suggesting that TM5b is a potential therapeutic target for NDI [5,37,38,62]. Knockdown of the gene encoding TM5b corrects the trafficking defect of the Ser256Ala AQP2 mutant. Specific inhibition of TM5b may be useful for both congenital and acquired NDI because the interaction between TM5b and phosphorylated AQP2 is critical for the final step of AQP2 trafficking.

Suga et al. showed that viral delivery of AQP2 in a lithium-induced rat model of NDI led to a reduction in urine output and an increase in urine osmolality, an effect that was limited to several days [111].

The ADP-activated purinergic P2Y12 receptor is an inhibitory GPCR that decreases intracellular cAMP levels upon activation. Zhang et al. showed that clopidogrel, a P2Y12 inhibitor, ameliorated lithium-induced polyuria, improved urine concentrating ability and AQP2 protein abundance, but not urine concentration in vasopressin-lacking Brattleboro rats [112].

Several studies show that statin induces apical accumulation of AQP2 in cultured cells and animal models. Simvastatin has been shown to induce membrane accumulation of AQP2 as a result of reduced clathrin-mediated endocytosis in LLCPK-1 cells [113]. Procino et al. showed that simvastatin increases AQP2 urinary excretion and urine osmolality in hypercholesterolemic patients [114]. In contrast, in healthy volunteers, urine osmolality at the start of a water loading test was lower on the day after simvastatin compared to the absence of simvastatin (760 vs. 388 mOsm/kg, *p* = 0.02) [115]. Despite this, the lowest urine osmolality increased modestly after the use of simvastatin (70 mOsm/kg to 85 mOsm/kg, *p* = 0.05). A double-blind, randomized, placebo-controlled pilot trial of atorvastatin for NDI in lithium users showed that atorvastatin (20 mg/d) did not significantly improve urinary osmolality compared to placebo over a 12-week period [116].

Metformin is a stimulator of 5′-AMP-activated protein kinase (AMPK). Efe et al. showed that metformin increases membrane accumulation of AQP2 and urinary concentration in V2R KO mice and a rat model of tolvaptan-induced NDI [117]. However, a clinical trial to examine the effect of metformin on congenital NDI patients was terminated due to lack of effect; however, it should be noted that only two patients were enrolled for this trial (ClinicalTrials.gov: NCT02460354).

MicroRNA (miRNA) inhibits the translation of target mRNA. Kim et al. report that two AQP2-targeting miRNAs, miR-32 and miR-137, decrease AQP2 expression in kidney collecting duct cells [118]. Ranieri et al. find that calcium-sensing receptor (CaSR) signaling reduces AQP2 abundance via miRNA-137 [119]. In mice, ablation of Dicer, which is required for miRNA maturation, induces NDI [120]. These findings implicate AQP2-targeting miRNAs as a therapeutic target in water balance disorders, including both dehydration and water retention.

Bogum et al. screened 17,700 small molecules in a cell-based assay and identified fluconazole as a candidate inhibitor of AQP2 trafficking [121]. Vukićević et al. show that fluconazole increased apical membrane localization of AQP2 caused by phosphorylation and ubiquitination of AQP2, and inhibition of RhoA [122]. Fluconazole also reduced urinary output in tolvaptan-treated mice.

Ando et al. found that the Wnt5a/calcium/calmodulin/calcineurin signaling pathway induced phosphorylation, trafficking, and expression of AQP2 [29]. Wnt5a successfully increased the apical membrane localization of AQP2 and urine osmolality in an NDI mouse model. In addition, the authors showed that arachidonic acid, which activates calcineurin by mimicking calmodulin, exerts similar effects on AQP2. Thus, calcineurin activators appear to be potential therapeutic targets for heritable NDI.

AKAP and PKA coordinate AQP2 regulation as described above. Ando et al. show that AKAP-PKA disruptors, which dissociate the binding of AKAP and PKA R subunits, increased PKA activity and contributed to AQP2 phosphorylation, trafficking, and water reabsorption. The low molecular weight compound 3,3′-diamino-4,4′-dihydroxydiphenylmethane (FMP-API-1) and its derivatives increase AQP2 activity to the same extent as vasopressin. Thus, AKAP-PKA disruptors are also a novel category of potential therapeutic drugs for NDI [123].

## 15. Vasopressin V2 Antagonists as Current Treatments for Water Retention, and Future Strategies for Pure Aquaretics by Direct AQP2 Inhibition

AQP2 also plays a critical role in water retention disorders such as heart failure, hepatic cirrhosis and SIADH. Vasopressin V2 antagonists (vaptans) are effective agents for water retention and hyponatremia by inducing free water diuresis as described above. Vaptans bind the V2 receptor, block downstream signaling and decrease the amount of AQP2 on the luminal membrane. Tovaptan has been shown to improve several heart failure symptoms in the short term; however, there is no effect on long-term prognosis [81,124]. There are several studies evaluating long-term V2 receptor antagonist therapy in chronic hyponatremia [125,126]. While these studies showed that serum sodium increased, there are no studies showing improvement in ‘hard’ outcomes such as hospitalization, morbidity, and mortality. One possible reason is that the decongestive efficacy of tolvaptan decreases with prolonged treatment. Moreover, tolvaptan is a relatively selective inhibitor of V2 activation that does not prevent activation of V1a and its potentially adverse cardiac and vascular effects. Pecavaptan is a dual V1a/V2 receptor antagonist that increases cardiac output and peripheral resistance in animal models, effects that are not observed using tolvaptan [127]. A clinical trial examining the effect of pecavaptan on clinical outcomes is ongoing (ClinicalTrials.gov: NCT03901729). However, vasopressin receptors (V1a, V1b, V2) have many downstream effects other than AQP2 function. The development of drugs that directly inhibit AQP2 can be expected to act as “pure aquaretics” that are highly specific and effective for water retention disorders.

## 16. Conclusions

AQP2 is a key molecule for water balance disorders. There are currently no drugs for the treatment of NDI that causes severe dehydration in the body. On the other hand, water retention and hyponatremia caused by excessive activation of AQP2 are often difficult to manage and worsen the prognosis of patients with heart failure and hepatic cirrhosis. The development of drugs targeting AQP2 is a research field that holds promise for achieving effective treatment of water balance disorders.

## Figures and Tables

**Figure 1 ijms-22-12950-f001:**
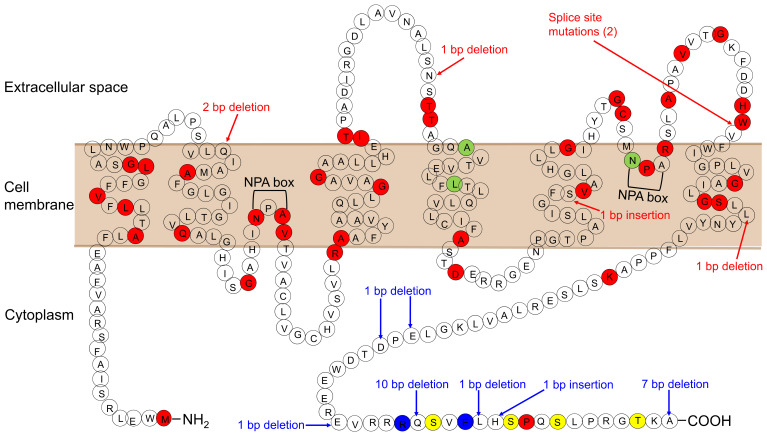
AQP2 mutations causing NDI. Mutations causing the autosomal-recessive form are shown in red. Mutations causing the autosomal-dominant form are shown in blue. Mutations, whose inheritance pattern is unknown, are shown in light green. Phosphorylation sites are shown in yellow.

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
