# Peer review of "Updates and Perspectives on Aquaporin-2 and Water Balance Disorders"

_ijms, 2021, doi:10.3390/ijms222312950_

Round 1
Reviewer 1 Report
Dear Authors,
This is an excellent review, that is well organized and appropriately summarizes recent studies.
Please check the figure number. There appears to be only one figure, so "Figure 2" which shows mutations causing NDI should be Figure 1.
Thank you very much.
Author Response
I am grateful to Reviewer #1 for pointing out my mistake. I have changed it to “Figure 1”.
Reviewer 2 Report
Dear Editor,
The manuscript by Noda et al. reviews the progress on the regulation of AQP2 under physiological and pathological conditions and discusses the upstream and downstream pathways involved in these regulatory mechanisms.
The review is comprehensive, informative, nicely-written, timely and up-to-date (in most parts). Authors were successful in providing some well compiled opinions and summaries. The inlcuded figures will be a good starting point for future studies and will be of interest for IJMS readers and beyond.
However, there is a number of major and minor points that would need to be addressed in order to improve the quality of this paper before it can be accepted for publication.
- This review overlooked some essential and up-to-date work regarding the recent advances in target validation and future therapies. I have made some suggestions below but authors are encouraged to consider citing updated references throughout the review, whenever possible.
-Authors are encouraged to highlight an important point to differentiate between expression vs. subcellular localisation for other but related examples of AQPs. For example, previous studies have shown an increased in AQP4 membrane localisation in primary human cells which wasn’t accompanied by a change in AQP4 protein expression levels. References:
https://www.ncbi.nlm.nih.gov/pubmed/31242419
https://pubmed.ncbi.nlm.nih.gov/31242419/
-Authors omitting a key study from 2020, demonstrating that targeting signalling mechanisms and phosphorylation is a viable therapeutic option. That study shows that CNS edema is associated with increases in total subcellular translocation following a phosphorylation event that involved calmodulin and PKA. Pharmacological inhibition of these signalling events prevents the development of CNS edema and promotes functional recovery in injured rats.
This role has been recently been confirmed by the work of Sylvain et al BBA 2021 which has demonstrated that targeting AQP4 effectively reduces cerebral edema during the early acute phase in in stroke using photothrombotic stroke model. They have also shown a link to brain energy metabolism as indicated by the increase of glycogen levels. Reference to be included:
https://www.cell.com/cell/fulltext/S0092-8674(20)30330-5.
https://pubmed.ncbi.nlm.nih.gov/33561476/
-The authors should review their discussion at the section of “Development of therapeutics for NDI by targeting AQP2 regulation” in light of the recent important publications highlighting the new approach of targeting regulation than the traditional “pore-blocking” approach for AQPs. References to be included:
https://pubmed.ncbi.nlm.nih.gov/34408336/
https://pubmed.ncbi.nlm.nih.gov/34499128/
Best.
Reviewer 3 Report
The review article by Yumi Noda and Sei Sasaki gives a thorough introduction to AQP2 and its roles played in water balance related disorders in human. In general, this article is well structured and nicely written. The usage of English language is with an acceptable standard.
There are some minor issues to be addressed:
- In line 14, the sentence “The key factor in the maintenance of body water balance is water reabsorption…….” is not precisely accurate. As there are other factors and mechanism important for maintaining water balance in the body, the beginning of the sentence should be “One of the key factors……”.
- In the abstract section between line 22 and 23, the sentences “Vasopressin is an upstream regulator of AQP2 and the vasopressin V2-receptor antagonist tolvaptan is effective and widely used for water retention with hyponatremia.” is not clear. Consider to re-write the sentences and provide more detailed and clear information.
- The sentence “Furthermore, it is expected that the development of drugs that directly target AQP2 may result in increased treatment specificity and effectiveness for water balance disorders.” in line 23-25 did not give the reason why the authors say this? Although I agree with this strategy, the authors should still compare this with the treatments targeting other factors such as V2R.
- Line 36-39, the authors only mentioned AQP2 translocation when the body is dehydrated, not mentioning the roles played by AVP and V2R in this situation. Please provide more logic information to the mechanism.
- The grammar of the sentence “with less phosphorylation of serine 264 and far less threonine 269” is a bit odd. Please re-write the sentence.
- It is already indicated in line 52 that cyclic adenosine monophosphate is abbreviated as cAMP. In line 118 and line 276, the “cyclic AMP” should be “cAMP”.
- Figure 1 is not clear, amino acids are in gray color, and the background is also in gray color. Please change the background color. The resolution of the figure seems to be poor. Consider to increase it.
- Please provide few conclusive sentences in the end of each section, and provide a “Conclusion” section to the article summarizing the information provided.
Round 2
Reviewer 2 Report
Dear Editor,
The authors have successfully addressed the majority of my comments and concerns in order to improve the quality of the manuscript.
I believe that the new sections, improved ones, and updated references, have contributed to enhancing the clarity of the manuscript, which I can now endorse for publication.
All the best!